# Quantitative mappings between symmetry and topology in solids

Zhida Song[1,2], Tiantian Zhang[1,2], Zhong Fang[1] & Chen Fang [ID] [1,3]

The study of spatial symmetries was accomplished during the last century and had greatly improved our understanding of the properties of solids. Nowadays, the symmetry data of any crystal can be readily extracted from standard first-principles calculation. On the other hand, the topological data (topological invariants), the defining quantities of nontrivial topological states, are in general considerably difficult to obtain, and this difficulty has critically slowed down the search for topological materials. Here we provide explicit and exhaustive mappings from symmetry data to topological data for arbitrary gapped band structure in the presence of time-reversal symmetry and any one of the 230 space groups. The mappings are completed using the theoretical tools of layer construction and symmetry-based indicators. With these results, finding topological invariants in any given gapped band structure reduces to a simple search in the mapping tables provided.

[1] Beijing National Research Center for Condensed Matter Physics, and Institute of Physics, Chinese Academy of Sciences, Beijing 100190, China. [2] University of Chinese Academy of Sciences, Beijing 100049, China. [3] CAS Center for Excellence in Topological Quantum Computation, Beijing 100190, China. Correspondence and requests for materials should be addressed to C.F. (email: cfang@iphy.ac.cn)

D istinct phases do not always differ from each other in their symmetries as expected in the Ginzburg-Landau paradigm. Two gapped phases having the same symmetry may be distinguished by a set of global quantum numbers called topological invariants[1–8]. These invariants are quantized numbers, whose types (integer, Boolean, and others) only depend on the symmetry group and the dimension of the system[9,10]. The invariants fully characterize topological properties that are unchanged under arbitrary adiabatic tuning of the Hamiltonian that preserves the relevant symmetry group. Materials having non-zero topological invariants are loosely called topological materials (whereas the technical term is symmetry protected topological states[11–13]), a new kind of quantum matter that hosts intriguing physical observables such as quantum anomaly on their boundaries[14–16], and are considered candidate materials for new quantum devices[17–21]. Success in finding these materials largely depends on the numerical evaluation (prediction) of the topological invariants in a given candidate material. However, even for electronic materials having weak electron correlation, where the topological invariants are best understood and expressed in terms of the wave functions of the valence bands, these calculations still prove quite challenging. In fact, numerically finding a new topological material has proved so hard that a single success[17,22–25] would have triggered enormous interest[26–32].

On the other hand, mathematicians and physicists have since long developed, via the representation theory of space groups, a complete toolkit for the study of the symmetry properties of bands in solids[33,34]. Given any point in momentum space, each energy level corresponds to an irreducible representation (irreps) of the little group at that momentum, depending on the Blöch wave functions at the level. Modern implementations of the density functional theory output both the energy levels and their Blöch wave functions for any given crystal momenta, such that finding the irreps of all valence bands in a band structure (BS) is now considered a solved problem that can be automated.

It has been eagerly hoped that quantitative relations exist between the topological invariants and the irreps in the valence bands at high-symmetry points in the Brillouin zone, i.e., the symmetry data of valence bands. These relations, if existed, would reduce the difficult task of finding the former to a routine calculation of the latter. However, the examples are rare[35–38]. Fu-Kane formula[35] for topological insulators protected by time-reversal symmetry (TI for short from now) is an exemplary one, mapping the four topological $\mathbb{Z}_2$-invariants to inversion eigenvalues at eight high-symmetry points. This simple golden rule considerably expedited the search for TI in all centrosymmetric materials via first-principles numerics[22]. Nevertheless, for general topological states in three dimensions protected by any one of the 230 space groups with and without time-reversal, or topological crystalline insulators[39] (TCIs), explicit formulae relating their topological invariants to symmetry data have so far been missing.

Recently, a solid step along this direction is made in refs. [40–43], where the authors systematically study the connectivity of bands in a general gapped BS and identify the constraints on the symmetry data in the form of linear equations called the compatibility equations. ref. [41] explicitly provides these relations for each space group and observes that if a symmetry data satisfying all compatibility relations cannot decompose into elementary band representations (sets of symmetry data of atomic insulators (AIs), given in the same paper), the material must be topologically nontrivial. ref. [40] shows that the symmetry data of any gapped BS can be compressed into a set of up to four $\mathbb{Z}_{n=2,3,4,6,8,12}$ numbers called symmetry-based indicators (SIs) (see Methods section for a brief review of SI). The set of SI is a lossless compression of symmetry data as far as topological invariants are concerned: all

topological invariants that may be extracted from symmetry data can be inferred from the corresponding SI. The theory presented in ref. [40] does not, however, relate SI to the topological invariants, the defining quantities of topological states: a BS having non-zero SI is necessarily topological, but the type of the topology in terms of invariants is unknown. The explicit expressions of the SI in terms of symmetry data are also missing in ref. [40].

This paper aims to complete the mapping between symmetry data and topological invariants in systems with time-reversal symmetry and significant spin–orbital coupling (the symplectic Wigner–Dyson class or class AII in the Altland-Zirnbauer system[44]). To achieve this, we first derive the explicit expression of each SI in all space groups (Supplementary Tables 1–3) and then, given any non-zero set of SI in every space group, we enumerate all possible combinations of topological invariants that are compatible with the SI (Supplementary Tables 4–8). These invariants include: three weak topological invariants $\delta_{w,i=1,2,3}$ [8], mirror Chern number $C_m$[23,45], glide plane (hourglass) invariant $\delta_h$[24], rotation invariant $\delta_r$[21,46,47], the inversion invariant $\delta_i$[21,36,37], a new $\mathbb{Z}_2$ topological invariant protected by screw rotations $\delta_s$, and finally a new $\mathbb{Z}_2$ topological invariant protected by $S_4$-symmetry $\delta_{S_4}$. The last two invariants are theoretically established in Supplementary Note 1. In the main results, the strong time-reversal invariant $\delta_t$[8] is assumed to vanish, so that the results are restricted to TCI only, or states that can be adiabatically brought to AIs in the absence of crystalline symmetries; the $\delta_t = 1$ cases are briefly discussed in the end of the Results section. The exhaustive enumeration maps 478 sets of SI to 3133 linearly independent combinations of topological invariants, as tabulated in Supplementary Table 7. A guide for reading this table is offered in Supplementary Note 7.

## Results

**An example showing the usage of our results**. Before entering into the derivation of the results, we use tin telluride (SnTe) crystal having space group $Fm\bar{3}m$ (#225) to illustrate how the results should be used in Fig. 1. One should first compute the symmetry data of the material, finding the numbers of appearances for each irrep in the valence bands at the high-symmetry momenta, namely, Γ, X, L, and W. This can be done in any modern implementation of first-principles numerics and here we use Vienna ab-initio simulation package[48,49]. From the symmetry data obtained in the top of Fig. 1, we apply the formulae given in Supplementary Tables 1 and 2 to find the SI, which in this case is a single $\mathbb{Z}_8$ number, and we find $z_8 = 4$. After this, we can use Supplementary Table 7 and find that $z_8 = 4$ corresponds to two and only two possible sets of topological invariants shown on the bottom of Fig. 1: it either has non-zero mirror Chern number $C_{m(001)} = 4$ (mod 8) for the $k_z = 0$ plane (and symmetry partners) or has mirror Chern number $C_{m(110)} = 2$ (mod 8) for the $k_x + k_y = 0$ plane (and symmetry partners). It is impossible, however, to distinguish these two cases using symmetry data, but advanced tools such as Wilson loops must be invoked. Further analysis shows that the latter state appears in the real material[23].

**Layer construction as an general approach**. A remarkable feature of all known TCIs is that any TCI can be adiabatically (without gap closing) and symmetrically tuned into a simple product state of decoupled, identical layers in real space, each of which decorated with some two-dimensional (2D) topological state[21,50–54]. This form of fixed-point wave function for a TCI is called its layer construction (LC). An analogy to AIs can be drawn to help understand the physical nature of LC in the following aspects: although an AI is built from decoupled point-like atoms, the building blocks of an LC are decoupled layers. Each atom in

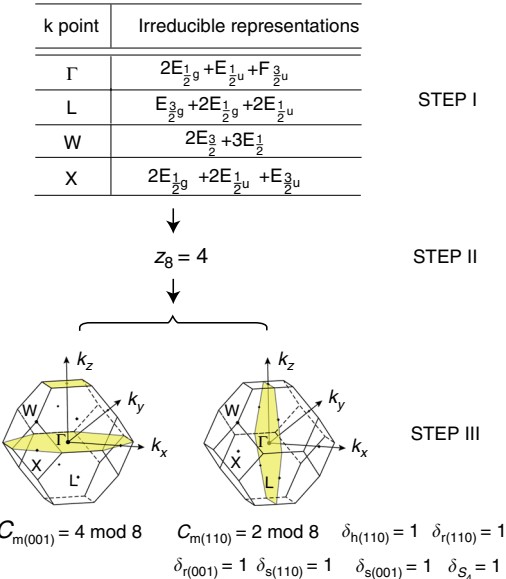

| k point | Irreducible representations | |
|---|---|---|
| $\Gamma$ | $2E_{\frac{1}{2}g} + E_{\frac{1}{2}u} + F_{\frac{3}{2}u}$ | STEP I |
| L | $E_{\frac{3}{2}g} + 2E_{\frac{1}{2}g} + 2E_{\frac{1}{2}u}$ | |
| W | $2E_{\frac{3}{2}} + 3E_{\frac{1}{2}}$ | |
| X | $2E_{\frac{1}{2}g} + 2E_{\frac{1}{2}u} + E_{\frac{3}{2}u}$ | |

$z_8 = 4$ STEP II

STEP III

$C_{m(001)} = 4 \bmod 8$ $C_{m(110)} = 2 \bmod 8$ $\delta_{h(110)} = 1$ $\delta_{r(110)} = 1$

$\delta_{r(001)} = 1$ $\delta_{s(110)} = 1$ $\delta_{s(001)} = 1$ $\delta_{S_4} = 1$

**Fig. 1** A demonstration of the diagnosis for tin telluride of space group $Fm\bar{3}m$ (#225) using our results. The table on the top shows the symmetry data obtained in the first-principles calculation (details given in text), where the numbers of appearance of each irrep in the valence bands are listed for each high-symmetry point in the face-centered-cubic Brillouin zone. From the data one finds the SI $z_8 = 4$ using Supplementary Tables 1 and 2, and then by searching for this indicator in Supplementary Table 7, two possible sets of topological invariants are found, listed at the bottom left and bottom right, respectively. The yellow planes in the Brillouin zone are where the mirror Chern numbers $C_{m(001)}$ and $C_{m(110)}$ are defined. Indices in the parentheses in subscript represent the directions of the corresponding symmetry elements. The real material has been shown in ref. [23] to have the topological invariants listed on the bottom right

an AI is decorated with electrons occupying certain atomic orbitals, whereas each layer is decorated with electrons forming a 2D topological state. The atomic orbitals of an atom in lattice correspond to the irreducible representations of the little group at that atomic position, whereas the possible topological states on a layer also depends on the little group leaving the layer invariant. In an LC, there are only two possible decorations: if the layer coincides with some mirror plane of the space group the state for decoration is a 2D mirror TCI with mirror Chern number $C_m$, and if it does not coincide it is a 2D TI. We define elementary LC (eLC) as an LC generated by a single layer in real space

$$(mnl; d) \equiv \{\mathbf{r} | (m\mathbf{b}_1 + n\mathbf{b}_2 + l\mathbf{b}_3) \cdot \mathbf{r} = 2\pi d \bmod 2\pi\} \quad (1)$$

Here $(mnl)$ are the Miller indices, and $\mathbf{b}_i$'s the reciprocal lattice vectors; generation here means we take all elements $g \in G$ to obtain the set of layers $E(mnl; d) \equiv \{g(mnl; d) | g \in G\}$ by acting $g$ on $(mnl; d)$. Every LC is a superposition of a finite number of eLCs and, thanks to the additive nature of all known topological invariants, the topological invariants of any LC is the sum of the invariants of all constituent eLCs.

For any space group, we exhaustively find all eLCs using the method detailed in Supplementary Note 3. Although the calculation of topological invariants is difficult for an arbitrary BS, it is easy for an eLC, thanks to its simple structure. In fact, the invariants only depend on how many times each symmetry element is occupied (see Supplementary Note 1 for proof, wherein the occupation for glide plane or screw axis is subtle). A symmetry element is a manifold in real space, where each point is invariant under some symmetry operation. It could be a discrete

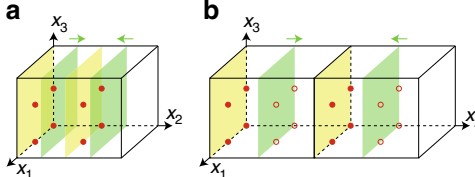

**Fig. 2** Layer constructions for space group $P\bar{1}$ (#2). **a** The yellow planes are (010; 0) and (010; $\frac{1}{2}$) respectively, and the two green planes are (010; $d$) and (010; $1-d$) with $d \neq 0, \frac{1}{2}$. The arrows mean that the two green planes can move towards each other without breaking inversion. **b** After doubling the unit cell along $x_2$-direction, the open dots are no longer inversion centers as they were, whereas the solid dots remain. Again the arrows mean that two green planes can move towards each other without breaking the inversion symmetry, after unit cell doubling

point such as an inversion center or a center of $S_4 : (x, y, z) \to (-y, x, -z)$, a line such as a rotation axis, or a plane such as a mirror plane. In this way, topological invariants for each eLC are calculated and tabulated in Supplementary Tables 5 and 6. On the other hand, the SI of an eLC are also easily calculated, detailed in Supplementary Note 5 again due to the decoupled nature of the layers. Matching the SI with invariants for each eLC, we hence find the full mapping between SI and topological invariants for TCI. For intuitive understanding, we also plot a set of figures (Supplementary Figs. 1–8) showing the invariants, SI, and phase transitions of eLCs.

**From indicators to invariants**. Here we take space group $P\bar{1}$ as an example to show the mapping between indicators and invariants and leave the general discussion in the Supplementary Notes 1 and 5. The space group $P\bar{1}$ has non-orthogonal lattice vectors $\mathbf{a}_{i=1,2,3}$ and inversion symmetry. Within a unit cell, there are eight inversion centers at $(x_1, x_2, x_3)/2$ in the basis of lattice vectors (the red solid circles in Fig. 2), where $x_i = 0, 1$. These inversion centers are denoted by $V_{x_1x_2x_3} \equiv (x_1, x_2, x_3)/2 \bmod 1$. A generic layer $(mnl; d)$ is given by $L = \{\mathbf{r} | (m\mathbf{b}_1 + n\mathbf{b}_2 + l\mathbf{b}_3) \cdot \mathbf{r} = 2\pi d \bmod 2\pi\}$, where $d \in [0, 1)$, and at least one of $m, n, l$ is odd (or they would have a common factor). If $d \neq 0, \frac{1}{2}$, we have $d \neq -d \bmod 1$, then under inversion a generated plane $L' = (mnl; 1 - d) \neq L$ is a different plane symmetric to $L$ about the origin. In that case, the two planes $L$ and $L'$ can adiabatically move towards each other without breaking any symmetry until they coincide, a process illustrated in Fig. 2a. The state decorated on $L$ and $L'$ are 2D TIs, and due to the $\mathbb{Z}_2$-nature, when $L$ and $L'$ coincide, the resultant double layer becomes topologically trivial. The eLC generated by $(mnl; d \neq 0, \frac{1}{2})$ is hence a trivial insulator. For $d = 0, \frac{1}{2}$, $L$ is invariant under inversion, and always passes four of the eight inversion centers, that is, $V_i$'s that satisfy the equation $mx_1 + nx_2 + lx_3 = 2d \bmod 2$. For examples, if $(mnl; d) = (010; 0)$, then $V_{000,001,100,101}$ are on $L$ (the left yellow plane in Fig. 2a). As each layer is decorated with 2D TI, eLC$(mnl; d)$ $(d = 0, \frac{1}{2})$ is the familiar weak TI having weak invariants

$$\delta_{w,1} = m \bmod 2 \quad \delta_{w,2} = n \bmod 2 \quad \delta_{w,3} = l \bmod 2 \quad (2)$$

Now we turn to the inversion invariant $\delta_i$, which is a strong invariant robust against all inversion preserving perturbations. Let us consider a perturbation that doubles the periodicity in the $(mnl)$-direction, whereas preserving the inversion center at origin. After the doubling, four of the eight inversion centers satisfying $mx_1 + nx_2 + lx_3 = 1 \bmod 2$ are no longer inversion centers, so that the plane $(mnl, \frac{1}{2})$ after the doubling no longer passes through any inversion center, and the generated eLC by $(mnl, \frac{1}{2})$ can be

trivialized after the doubling by pairwise annihilating with its inversion partner. Therefore, eLC($mnl$, $\frac{1}{2}$) can be trivialized, while keeping the inversion symmetry about the origin, thus having $\delta_i = 0$. In Fig. 2b, we take ($mnl$) = (010) as an example and doubled the unit cell. We see that the four inversion centers marked by empty circles are not inversion centers in the new cell and the blue plane at (010; $\frac{1}{2}$) in the original cell becomes (010; $\frac{1}{4}$) in the new cell. The two blue planes can move and meet each other at the yellow plane, denoted by (010;$\frac{1}{2}$) in the new cell. The eLC generated by ($mnl$; 0)-plane, however, passes all eight inversion centers in the enlarged unit cell and cannot be trivialized without breaking inversion (yellow planes in Fig. 2b), so that the inversion invariant $\delta_i = 1$.

After finding the invariants for all possible eLCs, we turn to the SI for each eLC. The SI group of P$\bar{1}$ takes the form $\mathbb{Z}_2 \times \mathbb{Z}_2 \times \mathbb{Z}_2 \times \mathbb{Z}_4$, wherein the first three are the weak TI indicators $z_{2w,i=1,2,3}$ and the last one is the $z_4$ indicator. The calculation method is briefly described in Supplementary Note 5 and here we only give the results. For eLC($mnl$; 0) and eLC($mnl$; $\frac{1}{2}$), their values are found to be ($m$ mod 2, $n$ mod 2, $l$ mod 2, 2) and ($m$ mod 2, $n$ mod 2, $l$ mod 2, 0), respectively. For this space group, the mapping from SI set to topological invariants is therefore one-to-one: $z_{2w,i} = \delta_{w,i}$ and $z_4 = 2\delta_i$.

**Convention dependence of topological invariants**. A subtle but important remark is due at this point. There are always eight inversion centers in a unit cell in the presence of inversion symmetry, and when translation symmetry is broken, only one, two, or four of them remain. In the definition of inversion invariant $\delta_i$, one of the eight is chosen as the inversion center that remains upon translation breaking. In the above example, when the unit cell is doubled, the origin was chosen as the center that remains, but if we chose $V_{010} = (\frac{1}{2}, 0, 0)$, which is a completely valid choice, the four open circles in Fig. 2b are still inversion centers but the solid circles are not after the doubling. In that case, we would find that eLC($mnl$, $\frac{1}{2}$) has $\delta_i = 1$ but eLC($mnl$, 0) has $\delta_i = 0$. The inversion invariant $\delta_i$ hence depends on the convention which one of the eight inversion centers in the unit cell is chosen in the definition of $\delta_i$. However, when we superimpose the two eLCs into an LC that passes all eight inversion centers in a unit cell, the value of $\delta_i$ is independent of the choice of the inversion center, as all eight are occupied in this LC. We emphasize that only if this is the case can we hope to observe the physical properties, such as the characteristic boundary states associated with the bulk invariant $\delta_i$[21], because physical observables should not depend on the conventions.

Moreover, similarly, as detailed in Supplementary Note 2, the rotation (screw) invariant $\delta_r = 1$ ($\delta_s = 1$) is convention-independent if and only if each rotation (screw) axis in unit cell is occupied by the LC for $n/2$ mod $n$ times, where $n = 2, 4, 6$ is the order of the rotation (screw) axis. For the $S_4$ invariant $\delta_{S_4} = 1$ or the hourglass invariant $\delta_h = 1$ to be convention-independent, the LC should occupy each $S_4$ center or glide plane for an odd number of times. Invariants that are convention-independent are marked blue in Supplementary Tables 7 and 8.

**The one-to-many nature of the mapping**. In the example of space group P$\bar{1}$, the mappings between indicators and topological invariants are one-to-one. However, this is in fact the only space group where mappings are bijective. By definition, different sets of indicators must correspond to different sets of invariants, but multiple sets of invariants may correspond to the same set of indicators, i.e., the mapping from indicators to invariants is one-to-many.

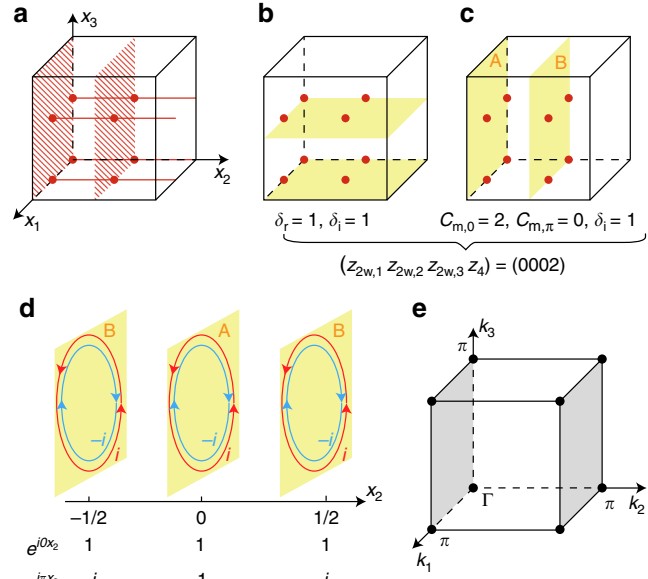

**Fig. 3** Two layer constructions for space group P2/m, sharing the same set of SI of (0002). **a** All symmetry elements of the space group in one unit cell, including eight inversion centers (red solid circles), four rotation axes (red solid lines), and two mirror planes (shaded planes). **b, c** LC1 and LC2 defined in the text, respectively. They have distinct topological invariants but identical indicators. **d** 3D Blöch wave functions in LC2 as superpositions of 2D Blöch wave functions with coefficients $e^{ik_2 x_2}$. Here we use red and blue loops to represent the 2D wave functions having mirror eigenvalues $i$ and $-i$, respectively, wherein $i$ wave functions have Chern number 1 and $-i$ wave functions have Chern number $-1$. For A-eLC the 3D Blöch wave functions at $k_2 = 0$ and $k_2 = \pi$ have the same mirror eigenvalues, leading to identical mirror Chern numbers at $k_2 = 0$ and $k_2 = \pi$. Although for B-eLC the Blöch wave functions at $k_2 = 0$ and $k_2 = \pi$ have opposite mirror eigenvalues, leading to opposite mirror Chern numbers at $k_2 = 0$ and $k_2 = \pi$. **e** The two mirror-invariant planes (gray planes) in Brillouin zone

To understand the one-to-many nature of the mapping more concretely, we look at the specific group P2/m, containing two mirror planes, four $C_2$-axes and eight inversion centers in each unit cell, all marked in Fig. 3a. Now we consider two different LCs illustrated in Fig. 3b, c: in Fig. 3b for LC1, two horizontal planes, each decorated with a 2D TI, occupy all four $C_2$-axes and all eight inversion centers, and in Fig. 3c for LC2, two vertical planes, each decorated with a mirror Chern insulator with $C_m = 1$, occupy the two mirror planes and the eight inversion centers.

As all inversion centers are occupied in LC1 and LC2, in both cases we have $\delta_i = 1$. LC1 occupies all four $C_2$-rotation axes, once each, thus having nontrivial rotation invariant $\delta_r = 1$, whereas LC2 does not occupy any of the rotation axes, having $\delta_r = 0$. On the other hand, LC2 occupies the two mirror planes, each with 2D TCI having $C_m = 1$. According to the calculation in Method section LC2 has mirror Chern numbers $C_m = 2$ at $k_z = 0$ plane and $C_m = 0$ at $k_z = \pi$ plane; LC1, not occupying any mirror plane, has vanishing mirror Chern number. LC1 and LC2 are therefore topologically distinct states.

Now we turn to the SI of LC1 and LC2. For space group P2/m, the SI have the same group structure $\mathbb{Z}_2 \times \mathbb{Z}_2 \times \mathbb{Z}_2 \times \mathbb{Z}_4$ as that of its subgroup P$\bar{1}$. In this case, the value of each indicator remains the same as we break the symmetry down to P$\bar{1}$. Viewed as LC in P$\bar{1}$, both LC1 and LC2 are the superpositions of eLC($mnl$, 0) and eLC($mnl$, $\frac{1}{2}$), thus having, by the additivity of SI, $z_{2w,i} = 0$ and $z_4 = 2$.

The failure in distinguishing LC1 and LC2 by indicators reveals a general ambiguity in the diagnosis of mirror Chern numbers. We can add additional even number ($2p$) of A-eLC and even number ($2q$) of B-eLC to LC1 such that the composite state has the same SI and $\delta_r$ with LC1 but non-zero mirror Chern numbers $C_{m,0} = 2p + 2q$, $C_{m,\pi} = 2p - 2q$. On the other hand, we can also add these additional eLCs to LC2 to get a state having the same SI with LC2 but different mirror Chern numbers $C_{m,0} = 2 + 2p + 2q$, $C_{m,\pi} = 2p - 2q$. The proof here can be generalized to any space group having mirror planes and perpendicular rotation axes, providing that the order of the rotation is even. In all these space groups, as shown in Supplementary Table 7, the TCIs having invariants $\delta_r = 1$ and $C_{m,0} - C_{m,\pi} = 0$ mod $2n$ and the TCIs having $\delta_r = 0$ and $C_{m,0} - C_{m,\pi} = n$ mod $2n$ have the same SI, here $n$ is the order of the rotation axis. The two possible sets of invariants shown in Fig. 1, wherein one is $C_{m(001)} = 4$ mod 8, $\delta_{r(001)} = 0$ and the other is $C_{m(001)} = 0$ mod 8, $\delta_{r(001)} = 1$, are the examples for $n = 4$.

**Indicators for time-reversal topological insulators**. In the SI group of each space group, except #174 and #187–190, there is one special indicator of $\mathbb{Z}_{2,4,8,12}$-type, denoted by $z_t$, marked red in Supplementary Tables 2 and 3. When this special indicator is odd, the system is the well-known three-dimensional (3D) time-reversal topological insulator[40] (TI for short). The essential difference between a TI and a TCI is that the former only requires time-reversal symmetry, such that it remains nontrivial even when all crystalline symmetries are broken. TI does not have LCs, so that the method we use does not apply to SI having $z_t \in$ odd. To construct states having $z_t \in$ odd, we first notice that TI is consistent with all space groups, such that for each space group, we have at least one state that is a TI. Then we can superimpose this TI with all existing LCs obtained, and generate gapped states for all non-zero combinations of SI with $z_t \in$ odd (but with five exceptions discussed in Discussion section).

Regarding $z_t \in$ odd, we comment that the value of $z_t$ generally has a convention dependence on the overall signs in the definition of inversion and the rotation operators. For example, in space group $P\bar{1}$, the defining properties of the symmetry operators are $\hat{P}^2 = 1$, $\hat{T}^2 = -1$ and $[\hat{P}, \hat{T}] = 0$. It is easy to check that the overall sign in front of $\hat{P}$ can be freely chosen without violating any of the above relations. In other words, without external references, it is unknown a priori if, e.g., an $s$-orbital should be assigned with positive or negative parity. Upon redefining $\hat{P} \rightarrow -\hat{P}$, a state having $z_4 = 1$ goes to $z_4 = 3$ and vice versa. In similar ways, it is proved in Supplementary Note 4 that states having the $\mathbb{Z}_8$-indicator $z_8 = 1, 3, 5, 7$ differ only by convention and so do the states have $\mathbb{Z}_{12}$-indicator $z_{12} = 1, 5, 7, 11$. In these cases, the convention refers to the overall sign in front of inversion operator and the sign in front of the rotation operator. It is difficult to distinguish these states from each other experimentally. However, here we emphasize that the SI $z_4 = 1, 3$ (so do the SI $z_8 = 1, 3, 5, 7$ and $z_{12} = 1, 5, 7, 11$) have a relevant difference under a fixed convention, which can be detected by the anomalous boundary between the two phases. For example, suppose we have a spherical sample of $z_4 = 3$ phase and fill the space outside the sphere with $z_4 = 1$ phase, then as long as the geometry keeps inversion symmetry, the boundary state on the spherical surface should be identical with the boundary state between $z_4 = 2$ and $z_4 = 0$ phases, which is known as one-dimensional helical mode (see Supplementary Note 1 for details). This is because we can deduct a background of $z_4 = 1$ phase both inside and outside the sphere without changing the boundary state.

The space groups #174, #187, #189, and #188, #190, where one cannot diagnose TI from SI, have the SI groups $\mathbb{Z}_3 \times \mathbb{Z}_3$ and $\mathbb{Z}_3$,

respectively, and the corresponding SI $z_{3m,0}$ and $z_{3m,\pi}$ are the mirror Chern numbers (mod 3) in the $k_3 = 0$ and $k_3 = \pi$ planes. In #188 and #190 $z_{3m,\pi}$ is trivialized by nonsymmorphic symmetry and thus the corresponding SI groups reduce to $\mathbb{Z}_3$. In these space groups, the TI invariant is the parity of $C_{m,0} - C_{m,\pi}$[23] whereas SI have ambiguity for the parities of mirror Chern numbers; thus, TI can never be diagnosed from SI. For example, $z_{3m,0} = 1$, $z_{3m,\pi} = 0$ can correspond to $C_{m,0} = 1$, $C_{m,\pi} = 0$ (a TI), or $C_{m,0} = -2$, $C_{m,\pi} = 0$ (not a TI).

## Discussion

A byproduct of this study is a complete set of TCIs that can be layer-constructed in all 230 space groups (Supplementary Tables 5–8), even including groups not having SI. The abundance of the states thus obtained naturally suggests the question: are all TCI states exhausted in these layer constructions? We regret to answer it in the negative: LC cannot give us the weak topological insulator states in five space groups, namely #48, #86, #134, #201, and #224. In any one of the five, there is a weak indicator $z_{2w}$, but all layer-constructed states have $z_{2w} = 0$. A common character of these space groups is that they have three perpendicular glide planes $\{m_{001} | \frac{1}{2}\frac{1}{2}0\}$ $\{m_{010} | \frac{1}{2}0\frac{1}{2}\}$ $\{m_{100} | 0\frac{1}{2}\frac{1}{2}\}$ such that any single layer having weak index $z_{2w,i} = 1$ would be doubled along the $i$-th direction and so the generated eLC has vanishing weak index. Explicit (non-LC) tight-binding models for the $z_{2w} = 1$ states are given in Supplementary Note 6, completing the proof that for any non-zero SI there is at least one corresponding gapped topological state. These corner cases are somewhat surprising as weak TI have so far been considered most akin to stacking of decoupled 2D TI.

Finally, we comment that all LCs can be used to build 3D symmetry protected topological states of bosons and fermions protected by space group $G$ plus a local group $G_L$. To do this one only needs to decorate each layer with a 2D SPT protected by $G_L$ instead of the 2D TI.

Towards the completion of the work, we have been aware of a similar study[55]. To our knowledge, the results, when overlapping, are consistent with each other.

## Methods

**A short review of SIs**. For each momentum in the Brillouin zone, there is an associate subgroup, called the little group, of the space group $G$, under the action of which the momentum is invariant up to a reciprocal lattice vector. A point is a high-symmetry point, denoted $K_j$, if its little group is greater than the little group of any point in the neighborhood. A fundamental theorem is that each band at momentum $K_i$ or multiplet of degenerate bands corresponds to an irreducible representation of the little group at $K_i$. The symmetry data of a BS is defined as the an integer vector $\mathbf{n}$, each element of which, $n\left(\xi_i^{K_j}\right)$, is the number of appearance of the $i$-th irreducible representation in the valence bands at the $j$-th high-symmetry momentum $K_j$, where $i = 1, ..., r_j$ labels the irreducible representations of the little group at $K_j$. One could further define the addition of two symmetry data as the addition of each entry, which corresponds to, physically, the superposition of two BSs.

For a gapped BS, the elements of its symmetry data cannot take arbitrary integers and there are constraints on the symmetry data known compatibility relations[40,41,43]. For example, gapped-ness requires that the occupation numbers at each $K_i$ be the same, i.e., $\sum_i n\left(\xi_i^{K_j}\right) = $ const. All compatibility relations are linear equations so that the symmetry data satisfying all these relations again form a smaller linear space, termed the BS space, denoted {BS}.

On the other hand, we consider the symmetry data of AIs. In AIs, the bands are generated by decoupled atomic orbitals placed at certain Wyckoff positions in the unit cell. By this definition, one finds that the symmetry data of AIs also form a linear space, denoted {AI} (also called the space of band representations[41]). Obviously a symmetry data $\mathbf{n} \in$ {AI} satisfies all compatibility relations, so {AI} $\subseteq$ {BS}. One then naturally considers the quotient space $X_{BS} = $ {BS}/{AI}. $X_{BS}$ is always a finite group generated by several $\mathbb{Z}_{n=2,3,4,6,8,12}$[40]. Each generator of $X_{BS}$ is called an SI.

The following properties of indicators should be mentioned: any two gapped BSs having different sets of SI must be topologically distinct, and any two different symmetry data having the same set of SI only differ from each other by the symmetry data of an AI.

In ref. [40], the authors calculate the group structure of the indicators for all 230 space groups. However, it does not give explicit formulae for the generators. In order for application, we derive all these formulae in Supplementary Note 4.

**Mirror Chern number of LC**. Below we explicitly calculate the mirror Chern numbers of LC1 and LC2 in Fig. 2. As shown in Fig. 3e, in BZ of space group $P2/m$ there are two mirror-invariant planes, i.e., the $k_2 = 0$ and $k_2 = \pi$ planes; thus, we have two mirror Chern numbers $C_{m,0}$ and $C_{m,\pi}$. We assume there are only two occupied bands in the vertical 2D TCIs in LC2 and denote the corresponding Blöch wave functions as $\left|\phi_{\pm i}(\mathbf{k}_{2D}, x_2)\right\rangle$. Here, $\pm i$ represent the mirror eigenvalues, where $i$ is the imaginary unit, $\mathbf{k}_{2D} = (k_1, k_3)$ is the 2D momentum, and $x_2$ is the position along $\mathbf{a}_2$ where the 2D TCIs are attached. We also assume that the wave functions with the mirror eigenvalue $i$ ($-i$) give a Chern number 1 ($-1$) such that the 2D mirror Chern number $C_m = 1$. Under the mirror operation $\hat{M}$ the 2D Blöch wave function $\left|\phi_{\pm i}(\mathbf{k}_{2D}, x_2)\right\rangle$ first get a mirror eigenvalue $\pm i$ and then move to the mirror position $-x_2$

$$\hat{M}\left|\phi_{\pm i}(\mathbf{k}_{2D}, x_2)\right\rangle = \pm i\left|\phi_{\pm i}(\mathbf{k}_{2D}, -x_2)\right\rangle \tag{3}$$

To calculate the mirror Chern numbers of LC2, we divide it into two subsystems: the eLC generated from A layer and the eLC generated from B layer (Fig. 3c). As the total mirror Chern numbers are the sum of mirror Chern numbers of the two subsystems, we need only to analyse the two subsystems, respectively. The 3D Blöch wave functions of A- and B-eLCs can be constructed as

$$\left|\psi^A_{\pm i}(\mathbf{k})\right\rangle = \sum_{x_2=0,\pm 1\cdots} e^{ik_2 x_2}\left|\phi_{\pm i}(\mathbf{k}_{2D}, x_2)\right\rangle \tag{4}$$

$$\left|\psi^B_{\pm i}(\mathbf{k})\right\rangle = \sum_{x_2=\pm\frac{1}{2},\pm\frac{3}{2}\cdots} e^{ik_2 x_2}\left|\phi_{\pm i}(\mathbf{k}_{2D}, x_2)\right\rangle \tag{5}$$

Due to Eq. (3), it is direct to show that $\left|\psi^A_i(k_1, 0, k_3)\right\rangle$ and $\left|\psi^A_i(k_1, \pi, k_3)\right\rangle$, both of which are superpositions of $\left|\phi_i(\mathbf{k}_{2D}, x_2)\right\rangle$ (Fig. 3d), have the same mirror eigenvalue $i$ (Fig. 3d). Thus, for A-eLC m the mirror Chern numbers at $k_2 = 0$ and $k_2 = \pi$ are all 1. On the other hand, $\left|\psi^B_i(k_1, 0, k_3)\right\rangle$ and $\left|\psi^B_i(k_1, \pi, k_3)\right\rangle$, again both of which have the Chern number 1, have mirror eigenvalues $i$ and $-i$, respectively (Fig. 3d). Thus, for B-eLC the mirror Chern numbers at $k_2 = 0$ and $k_2 = \pi$ are 1 and $-1$, respectively. Therefore, the total mirror Chern numbers in momentum space are $C_{m,0} = 2$ and $C_{m,\pi} = 0$ for LC2. It should be noticed that the values of $C_{m,0}$ and $C_{m,\pi}$ do not depend on the two band assumption we take: as long as the 2D TCI has $C_m = 1$, the results remain the same. On the other hand, the mirror Chern numbers of LC1 should be zero for both $k_2 = 0$ and $k_2 = \pi$ by the following argument. Without breaking mirror symmetry, each vertical plane can bend symmetrically towards the mirror plane until the two halves coincide on mirror-invariant planes in real space, due to the $\mathbb{Z}_2$-nature of each half, the folded plane is topologically equivalent to a trivial insulator. As LC1 can be smoothly trivialized without breaking mirror symmetry, it must have vanishing mirror Chern numbers.

## Data availability

The data and code that support the findings of this study are available from the corresponding author upon reasonable request.

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

## Acknowledgements

We acknowledge support from Ministry of Science and Technology of China under grant numbers 2016YFA0302400 and 2016YFA0300600, National Science Foundation of China under grant number 11674370 and 11421092, and from Chinese Academy of Sciences under grant number XXH13506-202.

## Author contributions

C.F. conceived the work and devised the main method to obtain the main results. Z.D.S. did the major part of the calculations. C.F. and Z.F. wrote the main text, whereas Z.D.S. and T.T.Z. wrote the Supplementary Information and plot all figures and tables.Data availabilityThe data and code that support the findings of this study are available from the corresponding author upon reasonable request.

## Additional information

**Competing interests:** The authors declare no competing interests.

