## [Peer Review File · Nature Communications]

Reviewers' comments:

Reviewer #1:

Searching new topological phases protected by crystalline space group symmetries is an important issue in the study of topological materials. To identify a topological phase, one need to compute the relevant topological invariant numerically, which is difficult and time-consuming in most cases. To overcome this difficulty, recently several groups of researchers have tried to find the relation between the symmetry data of band structure and the topological property of the system. In the present work, the authors have constructed an explicit mapping from the symmetry data to topological invariants using the layer construction method. They first derived the explicit expression of each symmetry-based indicator in terms of symmetry data, and then enumerated all possible combinations of topological invariants that are compatible with the symmetry-based indicator. In this way, the present work has provided an efficient way of computing the topological invariants of a material from first principles calculations.

Considering the tremendous impact of the Fu-Kane formula on the study of inversion-symmetric topological materials, I am sure that the present manuscript can also make a great contribution to the future study of new topological materials protected by crystalline symmetries. Although I think theoretical results shown in the paper are important and reliable, the presentation style of the manuscript should be improved further. Below I give several questions and comments which should be properly answered before I can make a decision on the paper.

Overall, section I, II, III are well-written whereas section IV, V, VI, VII are written too briefly so that it is very difficult to follow the discussion in Section IV-VII. I suggest the authors to expand those sections extensively by shifting related discussion presented in the Supplementary Materials to the main body. For instance,

(1) In Section IV, mirror Chern number is defined in both real and momentum spaces but their relation is not shown. Also related with the SI of LC1 and LC2 for space group P2/m, it is not clearly explained why they have the same SI. In fact, the concept and the properties of SI are not properly explained in the manuscript. So it is not easy to follow the related discussion unless the readers have enough background knowledge about it.

(2) In Section V, it is mentioned that space group #174 and #187-190 are exceptional. But there is no explanation about what makes those space groups distinct from other space groups.

(3) In Section VI, the authors distinguished two different types of Weyl semimetals. In one case, pairwise annihilation of Weyl points can occur without band inversion whereas pair-annihilation accompanies band inversion in the other case. But I do not understand what it means. As far as I know, in noncentrosymmetric systems, pair creation or pair annihilation of Weyl points can always happen whenever band inversion occurs at any point in the Brillouin zone.

(4) In Section VII, the authors mentioned another five space groups which are exceptional without explaining what makes those space groups to be exceptional, in what sense. Also, the authors have mentioned that even when some space groups do not have SI, layer construction predicts TCIs. Does it mean that there are TCIs which cannot be predictable based on SI? Why those TCIs are captured by layer construction?

(5) In the last paragraph in section I, I suggest to specify the space group number for SnTe since many tables are labeled by the space group numbers.

Reviewer #2 :

The main accomplishment of this work is a mapping between the symmetry data obtained by

examining symmetry irreps in momentum space and the topological invariants defining TCIs in systems with charge conservation and strong spin-orbit coupling (symplectic class). To perform this mapping, the authors use a layer construction to build the different TCIs and use their simple structure to calculate both the topological invariants and the symmetry indicators, thus establishing the mapping between the two. The work represents a very non-trivial extension of Ref. 40, which was restricted to classifying such symmetry data without providing explicit expressions for the symmetry-based indicators (SIs). By providing explicit expressions for the SIs and a quantitative mapping between those and the traditional topological invariants, this work has filled in the gaps left by previous works (especially Ref. 40) and in doing so, has significantly simplified the problem of computing topological invariants in materials and finding new ones. This makes the work, in my opinion, of great value to theoreticians and experimentalists working in the continuously growing field of topological materials. For these reasons, I recommend the work for publication provided the following comments are addressed:

1. The discussion of the one-to-many nature is, in my opinion, incomplete. In particular, the ambiguity in identifying integer invariants, e.g. mirror Chern number, using symmetry data is not addressed. Although this is something that is known for some time and that was discussed in Ref. 40, I think it has to be included in the discussion of Sec. IV. Also related to this is the statement “ $z_8 = 4$ corresponds to two and only two possible sets of topological invariants: it either has nonzero mirror Chern number $C_m(001) = 4$ for the $k_z = 0$ -plane, or has mirror Chern number $C_m(110) = 2$ for the $k_x + k_y = 0$ -plane” in the discussion of TnSe. This statement is not entirely correct since other possibilities, such as $C_m(001) = -4$ or 12 for the $k_z = 0$ -plane, are also consistent with the same SI.
2. In the discussion of the odd elements of z_s in Sec. V, the authors assert that the distinction between them is purely conventional. While this is true in an absolute sense, it is not true in a relative sense. For example, for the case of Z_4 , the phases corresponding $z_4=1$ and $z_4=3$ will add up to a completely trivial phase $z_4=0$, while doubling any of them leads to a non-trivial phase $z_4=2$. This means there is a relative distinction between the phases independent of the convention, since the “addition” of phases requires using the same convention for both.
3. The discussion of Sec VII and the related appendix is very brief. Despite providing an explicit tight-binding example in the appendix, the authors do not provide a clear reason why those five space groups are special. In addition, it is puzzling to me that the argument of Refs. 53-57 (which, to my knowledge, was restricted to point group symmetries) for the layer construction fails in this particular case while working for all the other cases. A revisiting of the layer construction argument (possibly in the Appendix) explaining in more detail when it is expected to fail will make this part much clearer.
4. This last point is optional, but it may be useful to have a brief discussion on the surface states corresponding to a given topological invariant. Such discussion will be very helpful when comparing to experiments which mostly probe topology by looking at surface states.

Reply to the First Reviewer:

We thank the Reviewer for the careful reading and evaluation of our work, and also for the generally positive assessment. The criticism on the presentation style is also appreciated and carefully addressed in our reply and/or revision. The paper was initially purported for another Nature journal that has rather strict word limit, so some parts were cut short than they should be in the main text. In the revision, we extend Sections IV, V and VII to expound and clarify the issues raised by the Reviewer. We are confident that all issues have been addressed in full. Below we reply to the issues in the order they are raised.

--(1) In Section IV, mirror Chern number is defined in both real and momentum spaces but their relation is not shown. Also related with the SI of LC1 and LC2 for space group P2/m, it is not clearly explained why they have the same SI. In fact, the concept and the properties of SI are not properly explained in the manuscript. So it is not easy to follow the related discussion unless the readers have enough background knowledge about it.

We agree that the concept of SI should be formally introduced and explained. This part is now added in the introduction where symmetry indicators are first mentioned. For simple Bravais lattices, if mirror reflection symmetry, then there are two inequivalent mirror invariant planes in one unit cell in real space at $z=0$ and $z=1/2$, and also two mirror invariant planes in momentum space at $kz=0$ and $kz=\pi$. The relations between the mirror Chern numbers in the real and in the momentum spaces are as follows: $MC(kz=0)=MC(z=0)+MC(z=1/2)$, $MC(kz=\pi)=MC(z=0)-MC(z=1/2)$. In revision, these relations are moved from the Supplementary Materials to the main text. The calculation of the indicators of LC1 and LC2 are also presented in some detail in the revision.

--(2) In Section V, it is mentioned that space group #174 and #187-190 are exceptional. But there is no explanation about what makes those space groups distinct from other space groups.

These space groups have indicators in the form of Z_3^k , where $k=1$ or 2 . Z_3 indicators are shown to indicate the mirror Chern numbers at $kz=0$ and $kz=\pi$, up to a multiple of 3. It is known that $MC(kz=0)-M(kz=\pi) \bmod 2$ equals the strong TI index, but since the Z_3 indicator could not distinguish $+1$ from -2 , or -1 from $+2$, they cannot give us $MC(kz=0)-M(kz=\pi) \bmod 2$. This discussion is now added in Section V.

--(3) In Section VI, the authors distinguished two different types of Weyl semimetals. In one case, pairwise annihilation of Weyl points can occur without band inversion whereas pair-annihilation accompanies band inversion in the other case. But I do not understand what it means. As far as I know, in noncentrosymmetric systems, pair creation or pair annihilation of Weyl points can always happen whenever band inversion occurs at any point in the Brillouin zone.

The discussion of Weyl points is now entirely removed from the paper, as we think the paper should be focused on the relations between symmetry eigenvalues and topological invariants in gapped band structures. Yet, we answer the Reviewer's question: if the number of the pairs of Weyl points

is not a multiple of the order of the little group at Gamma, then the creation/annihilation of some pairs must happen at high-symmetry momenta. For example, in the presence of C4 rotation, and one has six pairs of Weyl points, we immediately know that two pairs of them must be created/annihilated at non-generic momenta. One could happen at GZ and the other at MA, or two can both happen at XR (there are two XR's). If there are five pairs, then we know one pair has to come from either GZ or MA.

--(4) In Section VII, the authors mentioned another five space groups which are exceptional without explaining what makes those space groups to be exceptional, in what sense. Also, the authors have mentioned that even when some space groups do not have SI, layer construction predicts TCIs. Does it mean that there are TCIs which cannot be predictable based on SI? Why those TCIs are captured by layer construction?

There are two questions asked in this item. These five groups are special because in all other space groups, each nonzero combination of indicators corresponds to at least one TCI from layer construction. However, for these five groups, there is one Z2 weak indicator, but none of the layer constructions has Z2=1. It means that in these five groups, weak TI, whose existence is proved in the tight-binding model, cannot be layer-constructed. This is the first instance of a TCI that is non-layer-constructible. We do not know the underlying reason for the failure of the layer construction in these groups, and we leave it for future study. All that we know is that all five groups have #48 as common subgroup and #224 as common supergroup without enlarging the unit cell.

For the second question, there are TCIs not captured by SI diagnosis. In fact, only half of the space groups have indicators, but 227 out of 230 have nontrivial TCIs. The general eigenvalue diagnosis problem may be described as follows. A certain symmetry group S can protect a certain topological phase, or namely, protect some topological invariant Z_n . In almost all cases, the eigenvalues of S alone are not useful at all in diagnosing if a given band structure has nontrivial Z_n invariant. For example, C4-rotation protects a Z_2 -invariant, but this Z2 invariant has nothing to do, and so cannot be diagnosed using, the C4 eigenvalues of the bands. The diagnosis always requires a larger symmetry group $G > S$, and the eigenvalues of G sometimes have relations to the Z_n invariants protected by S. For TCI, S can be either of the following: mirror, (n=2,4,6)-fold rotation symmetry, (n=2,4,6)-fold screw axis and S_4 ; and G runs each one of the 230 space groups that contains S. The main result of the paper is that we exhaust the relations that exist between the eigenvalues of G and the invariants protected by S, for arbitrary combinations of S and G.

--(5) In the last paragraph in section I, I suggest to specify the space group number for SnTe since many tables are labeled by the space group numbers.

The space group number is 225 and has been added in the revision.

Reply to the Second Reviewer:

We thank the Reviewer for the very professional comments which cannot be expected from anyone but the top experts with similar interests. It is a treat reading these feedbacks, and fun replying them.

--1. The discussion of the one-to-many nature is, in my opinion, incomplete. In particular, the ambiguity in identifying integer invariants, e.g. mirror Chern number, using symmetry data is not addressed. Although this is something that is known for some time and that was discussed in Ref. 40, I think it has to be included in the discussion of Sec. IV. Also related to this is the statement $\chi_8 = 4$ corresponds to two and only two possible sets of topological invariants: it either has nonzero mirror Chern number $C_m(001) = 4$ for the $k_z = 0$ -plane, or has mirror Chern number $C_m(110) = 2$ for the $k_x + k_y = 0$ -plane in the discussion of TnSe. This statement is not entirely correct since other possibilities, such as $C_m(001) = -4$ or 12 for the $k_z = 0$ -plane, are also consistent with the same SI.

We agree with this point in full. In revision it is clearly stated that all mirror Chern numbers can only be determined up to $2n$, n being the order of the main axis, and that this is actually the universal ambiguity in all space groups. In revision, we use space group $P2/m$ as example to explain the ambiguity in the diagnosis of mirror Chern numbers in detail.

--2. In the discussion of the odd elements of z_s in Sec. V, the authors assert that the distinction between them is purely conventional. While this is true in an absolute sense, it is not true in a relative sense. For example, for the case of Z_4 , the phases corresponding $z_4=1$ and $z_4=3$ will add up to a completely trivial phase $z_4=0$, while doubling any of them leads to a non-trivial phase $z_4=2$. This means there is a relative distinction between the phases independent of the convention, since the "addition" of phases requires using the same convention for both.

We agree with this point in full. The phrase "purely conventional" verges on misleading, and we should just have plainly said that the value of the invariant depends the choice of different symmetry elements within a unit cell. Without changing the choice, the two phases of $z_4=1$ and $z_4=3$ are certainly distinct. In a thought experiment, one could make two concentric spheres, inner one being $z_4=1$ and outer $z_4=3$, and we expect the 1d helical mode protected on their interface just like the one on the surface of a $z_4=2$ TCI. These statements are added in the revision.

--3. The discussion of Sec VII and the related appendix is very brief. Despite providing an explicit tight-binding example in the appendix, the authors do not provide a clear reason why those five space groups are special. In addition, it is puzzling to me that the argument of Refs. 53-57 (which, to my knowledge, was restricted to point group symmetries) for the layer construction fails in this particular case while working for all the other cases. A revisiting of the layer construction argument (possibly in the Appendix) explaining in more detail when it is expected to fail will make this part much clearer.

The purpose of the tight-binding is to show that the TCI with $zw=1$ exists, and the fact that it cannot be layer-constructed makes this TCI the first instance (first five to be exact) where layer construction fails. The statement "the argument of Refs. 53-57 was restricted to point group symmetries" is erroneous, because in Ref.55 it is shown that glide plane TCI can also be layer-constructed. With these references and the current paper where screw axis TCI and S_4 TCI are also shown layer-constructible, we were confident in assuming that all TCIs can be layer constructed in all space groups. In light of these counterexamples, we now know that this

assumption is false. However, we honestly do not know the underlying reason, which gives us all known TCI phases, should fail for these space groups when $zw=1$. What we know is that, as we follow the our procedure enumerating all possible layer constructions, the space group symmetries always enforce an even number of layers along each direction in every unit cell, rendering a weak TI phase impossible. In revision, we extend Section VII explaining this failure in some detail, but still could not give any reason.

--4. This last point is optional, but it may be useful to have a brief discussion on the surface states corresponding to a given topological invariant. Such discussion will be very helpful when comparing to experiments which mostly probe topology by looking at surface states.

We choose to add a figure summarizing the surface states for each TCI with a single nonzero invariant. Thanks to the simple group structure of topological invariants, the surface state of TCI with multiple nonzero invariants is the superposition of the surface states corresponding to the constituent invariants, with the constraint that the surface state complies with the symmetry on the boundary. This is added in the Supplemental Materials.

REVIEWERS' COMMENTS:

Reviewer #1 (Remarks to the Author):

I have read the authors's response and the revised manuscript carefully. I think the authors have answered all my questions and comments appropriately. In particular, the presentation style of the manuscript has improved further so that the revised manuscript can be accessible to broad readers. I recommend the revised manuscript for publication in its present form.

Reviewer #2 (Remarks to the Author):

The authors have addressed all my previous comments satisfactorily. The presentation has improved significantly in the revised manuscript and the inaccurate/unclear points I pointed to in my earlier review were all clarified. I recommend for publication.

Addressing Referee's comments

In the last round of review, neither of the referees raised any issue, and both recommended publication